# Leveraging Signal Processing and Machine Learning for Automated Fault Detection in Wind Turbine Drivetrains

Faras Jamil[1,2], Cédric Peeters[1], Timothy Verstraeten[1,2], and Jan Helsen[1]

[1]Acoustics & Vibration Research Group / OWI-Lab, Vrije Universiteit Brussel
[2]Artificial Intelligence Lab, Vrije Universiteit Brussel

**Correspondence:** Faras Jamil (faras.jamil@vub.be)

**Abstract.** Wind energy is considered a sustainable renewable energy source; however, it faces the challenge of significant operating and maintenance costs. The research proposes a hybrid fault detection method to combine the physical domain knowledge with the machine learning models to provide an overview of the health of wind turbine drivetrain components. Signal processing indicators are computed from raw vibration signals measured from strategically placed accelerometers over drivetrain components. It produces an immense number of indicators as each indicator is sensitive towards certain types of faults, and manual monitoring becomes an unfeasible task. The machine learning models are trained using signal processing indicators and SCADA data. The normal behaviour modelling technique is employed to learn the healthy operation of the machine from data collected during healthy machine operation. The trained normal behaviour machine learning models label each indicator in a healthy or faulty state over time. The labelled state-of-the-art signal processing indicators are fused to provide a high-level health status overview of wind turbine drivetrain components. It helps to derive the required details from many condition indicators, which is valuable when managing multiple components in a single wind turbine across an entire wind farm. The proposed hybrid fault detection method is validated on an offshore wind farm with multiple years of condition monitoring data. It provides a high-level health overview that is readily understandable for non-expert wind farm operators, and for more detailed fault analysis, experts can conduct a comprehensive inspection.

## 1 Introduction

Renewable energy has experienced significant growth in recent years and has reduced the impact of global warming. In 2022, international investments in the renewable energy sector reached USD 1.3 trillion to decarbonise fossil-fuel-based energy production (ire, 2023). The growing interest in renewable energy has led to a substantial increase in clean, green energy production. The global installed wind energy capacity escalated to 906 GW due to the fast growth observed during recent years (Hutchinson and Zhao, 2023). The increasing interest in renewable and wind energy is accompanied by the challenge of significant operating and maintenance (O&M) costs. In the case of offshore wind, the O&M cost accounts for 30% of the total energy cost, primarily due to the remote and challenging environmental conditions of offshore locations. Offshore wind energy sites are advantageous for wind energy due to the availability of more consistent and strong winds to harvest (Gao and Odgaard, 2023). This situation offers ample opportunities for cost reduction in offshore wind energy by identifying faults at

early stages to plan efficient group maintenance strategies by combining multiple wind turbines or components (Wang et al., 2022b). It is crucial to accurately determine the health status of wind turbines to plan efficient maintenance strategies during low energy demand and suitable weather conditions to reduce the O&M cost (Helsen et al., 2017).

The advancement of Industry 4.0 introduced Internet of Things (IoT) devices, facilitating the transfer of sensor data over the Internet (Ma et al., 2022). The wind industry utilised this opportunity and equipped wind turbines with various IoT sensors to gather a wide range of data (Verstraeten et al., 2019a). Moreover, the drive to automate every conceivable industrial process has made the mechanical industry increasingly intricate, owing to the complex interrelations between numerous complicated components and procedures. Wind turbines are complex machines because they operate under constantly evolving operating conditions in challenging weather and environmental settings. The non-stationary conditions of wind turbines pose significant challenges to fault detection (Liu et al., 2023). As a result, there is a growing need for more advanced maintenance strategies to effectively manage and uphold wind turbine performance (Zonta et al., 2020). The big data collected from multiple IoT sensors enables continuous health monitoring of wind turbines. It transforms the maintenance strategies from periodic or reactive to predictive (Nejad et al., 2022). Currently, predictive maintenance is the most optimal maintenance strategy, which allows planning future group maintenance by identifying faults at the initial stage, leveraging the continuous data measurements provided by IoT sensors. Predictive maintenance methods involve training machine learning models using historical data from IoT sensors to predict the current health state of a wind turbine. It provides insight to execute group maintenance or arrange necessary measures to extend the wind turbine's operational life (Zhang et al., 2019; Carvalho et al., 2019). The life of a wind turbine affected by an early-stage fault can be increased by only utilising it during high demand (Verstraeten et al., 2019b). The wind energy O&M cost reduces when reliability is improved and wind turbines are available to produce energy in line with demand (Clark and DuPont, 2018).

The IoT sensors installed on wind turbines have the capacity to measure various types of data, primarily used for vibration analysis, acoustics, oil analysis, strain measurement, and thermography. Supervisory Control and Data Acquisition (SCADA) is a widely used data acquisition and monitoring system for wind turbines. It collects various types of data, including wind speed, power output, and rotor speed, which are essential for assessing the operating condition of the wind turbine. The measured data is used to monitor components such as gearboxes, generators, main bearings, blades, and towers (García Márquez et al., 2012). SCADA data has been utilized for detecting faults in the generator (Chesterman et al., 2022; Peter et al., 2022) and the main bearing (Beretta et al., 2021). SCADA data offers a cost-effective health monitoring solution since it eliminates the need for additional sensor installations. However, SCADA-based condition monitoring is unreliable; only a small subset of SCADA parameters is suitable for fault detection (Nejad et al., 2022). A large subset of SCADA parameters are utilized as machine learning model features to enhance fault detection and improve the reliability of SCADA-based condition monitoring (Renström et al., 2020; Lima et al., 2020; Dienst and Beseler, 2016). Among condition monitoring techniques, vibration analysis has emerged as a primary method for condition monitoring (Nejad et al., 2022; Helsen et al., 2017; Peeters et al., 2019a). However, raw data is not sufficient to provide insights to develop effective predictive maintenance strategies. Signal processing indicators derived from raw vibration data are used to monitor the health status of wind turbines. Experts are required to monitor individual indicators to identify emerging fault trends. Numerous signal processing methods are used to compute

such indicators, with each one sensitive to specific types of faults. Therefore, it is unfeasible for experts to continuously monitor each indicator, especially when a wind farm has multiple wind turbines, and each wind turbine contains many components. Machine learning models can provide a high-level health status (Jamil et al., 2023b, a), or label the indicators with healthy and alarm states for easy interpretation (Peeters et al., 2019b).

The signal processing indicators for fault detection are categorized into two groups: time domain and frequency domain indicators. Time domain indicators, such as mean and standard deviation, are statistical parameters that enable the determination of health degradation trends by monitoring deviations from established normal behaviour. These time domain features include statistical parameters such as root mean square, kurtosis, peak-to-peak, Moors kurtosis, peak energy index, and crest factor (Peeters et al., 2017, 2018a, 2019b). Time domain indicators significantly simplify health analysis since they do not require knowledge of the characteristic frequencies of components. Nonetheless, these indicators only indicate which sensor has detected a fault, without offering insights into the specific nature of the fault. In contrast, frequency-domain indicators not only detect faults but also specify which component is experiencing the fault by utilizing characteristic frequencies associated with those components. Bearing characteristic frequencies are the inner race, outer race, roller cage, and roller, while gears have distinct characteristic frequencies associated with gear meshing. These characteristic frequencies are tracked in the spectral domain of the signal and its envelope (HO and RANDALL, 2000; McCormick and Nandi, 1998). Fault detection methods based on cyclostationarity are gaining popularity in rotating machinery fault detection applications (Antoni, 2009). These methods highlight the modulation in signals introduced by faults in rotating components.

There is a rising interest among the research community in employing machine learning models for fault detection (Liu et al., 2018; Xiang et al., 2022). The most commonly used machine learning models include artificial neural networks (Marugán et al., 2018), support vector machines (Vidal et al., 2018; Widodo and Yang, 2007), and deep neural networks (Dibaj et al., 2023; Jia et al., 2016; Ibrahim et al., 2016). These machine learning applications are implemented by leveraging signal processing features as they depict substantial fault trends (Peeters et al., 2019b; Jamil et al., 2023b, a; Perez-Sanjines et al., 2023). However, a primary challenge for machine learning models is the scarcity of fault cases, as machines typically operate in a healthy state for the majority of their operational time. Therefore, training a classifier to distinguish between normal and faulty health states may not be an optimal technique. Transfer learning provides a viable solution to address the issue of limited data availability in the machine learning discipline. It enables the transfer of data or knowledge from similar domains to enhance the prediction performance of machine learning models (Zhuang et al., 2021). In the context of fault detection, transfer learning enhances detection capabilities by transferring learned knowledge from a similar source domain to a target domain (Bai et al., 2021), while avoiding negative transfer (Jamil et al., 2022). However, transfer learning does require a few known faulty cases to improve supervised learning fault detection models. It implies that in order to deploy a fault detection model in industry, a wind turbine must experience a failure event that provides the necessary data to train a fault detection classifier. The acquisition of fault data for machines poses a significant challenge, as it requires the occurrence of faults before training a robust health status classifier to differentiate between healthy and faulty states. However, healthy data is available to measure after the machine begins operations. A more practical approach in such cases involves employing a machine learning model capable of learning healthy behaviour and detecting faults by identifying deviations from the learned normal behaviour. The Normal

Behavior Model (NBM) is trained on the expected behaviour and identifies deviations as anomalies. This approach allows for the utilisation of healthy wind turbine data to train NBMs, and detect a potential fault when any deviation from the expected behavior is observed. (Peeters et al., 2019b; Helsen et al., 2018; Wang et al., 2022a). Peeters et al. (2019b) has introduced a hybrid fault detection method that leverages signal processing statistical indicators computed during a healthy operating period to train an NBM. The trained NBM is able to label indicators' healthy states and any possible deviations as a faulty trend. Temperature signals are less sensitive as compared to the vibration signals towards any fault introduction, however, they do exhibit trends that can indicate fault trends.NBMs trained on healthy temperature signals are able to successfully detect gearbox faults in wind turbines (Helsen et al., 2015, 2018). Chesterman et al. (2022) compares statistical and machine learning NBM pipelines to detect wind turbine generating bearing fault using SCADA data. Perez-Sanjines et al. (2023) employed coherence maps derived from vibrational signals to train normal behaviour deep learning models to detect mechanical faults in wind turbine rotating components. Normal behaviour modelling is the most effective machine learning approach for mechanical condition monitoring and failure prediction, as concluded from a comparison of various SCADA data-based condition monitoring techniques (Chesterman et al., 2023). The key advantages of Normal Behavior Modeling (NBM) include its ability to perform unsupervised learning and detect failures without prior exposure to them. NBM can be easily adapted to various types of data from any component or process by learning normality and triggering an alarm when the normality is violated.

The proposed method is a hybrid fault detection approach to provide a high-level health status summary by adapting to changing operating conditions by using vibration and SCADA data. It combines the computational capabilities of machine learning with domain knowledge of signal-processing indicators. This method leverages domain knowledge to train machine learning models, providing results that are both easily interpretable and explainable, compared to black-box machine learning models only predicting the presence of faults. The effectiveness of the proposed method has been validated across multiple real-life wind farms, spanning several years of operational data. It has demonstrated a notable capability to detect fault trends at their early stages. The key contributions of this research can be summarised as follows:

- Development of a fully automated vibration and SCADA data-based drivetrain monitoring pipeline.

- The method introduces physical domain knowledge to enhance machine learning models' prediction performance.

- It provides a high-level health status overview by adapting to changing operating conditions.

- It has been validated on many real-world wind farms, leveraging over multiple years of data, and exhibited promising results.

## 2 Hybrid condition monitoring fault detection method

The proposed approach is a hybrid condition monitoring method for fault detection. This method integrates vibration-based signal processing indicators and SCADA data with machine learning models. Although SCADA data has not been designed primarily for wind turbine condition monitoring (Tautz-Weinert and Watson, 2017), the proposed method derives operational

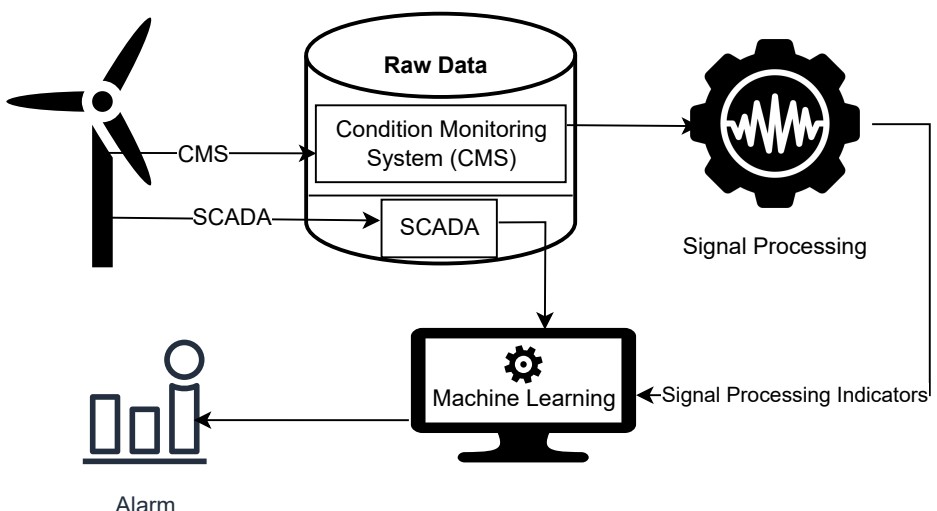

**Figure 1.** The hybrid condition monitoring pipeline commences with data acquisition from a wind turbine. This data is then used to compute signal processing features, which are subsequently passed as input into machine learning models alongside SCADA data to predict alarms.

information about the wind turbine from this data. A range of signal processing techniques are employed to compute indicators from vibration signals, including time-domain statistical indicators, frequency-domain spectral features, and cyclostationary indicators. These computed indicators serve as the input for machine learning NBMs trained to classify the indicators into healthy or faulty states. Subsequently, the approach calculates the count of indicators labelled as faulty at each time step to provide a high-level overview of the wind turbine's health. Figure 1 offers a visual representation of the hybrid condition monitoring pipeline, which commences with data acquisition from sensors installed on the wind turbine drivetrain. The SCADA data is collected through the SCADA system, while the vibrational data is measured by the condition monitoring system. Since these systems operate at different frequencies and intervals, the data from both sources are integrated by aligning them based on matching timestamps. The combined dataset is derived by merging the vibration data measurements with their corresponding SCADA data measurements. This unified dataset is passed to machine learning-based NBMs to label the measurements into healthy and faulty states. Finally, in the last stage, alarms are triggered based on the number of indicators classified as faulty per timestamp.

The proposed hybrid condition monitoring pipeline is structured into three distinct steps:

- Computation of signal processing time-domain and frequency-domain condition indicators

- Labelling computed indicators into healthy and faulty states using normal behaviour machine learning models

- Alarming to provide a high-level health status of the wind turbine

## 2.1 Signal processing

The condition indicators that are fed to the normal behaviour models are the result of an extensive signal pre-processing phase
that tries to track any significant changes in the measurements. The following paragraphs detail the different steps required to
arrive at a set of meaningful and effective set of condition indicators for fault detection.

Due to the non-stationary nature of wind turbine drivetrains, the measured vibration signals cannot be fully analyzed directly without knowledge of the rotating speed or instantaneous angular speed (IAS) of the drivetrain shafts. Given that not
every turbine has a high-resolution angle encoder installed on its drivetrain, the first step in the signal processing pipeline is
the automated estimation of the IAS directly from the vibration signals. To achieve this automated estimation, knowledge of
the kinematic orders present in the gearbox is necessary, i.e. the gear ratios. Using these kinematic orders, we can employ
the harmonic frequencies that each gear produces as input for the multi-order probabilistic approach (MOPA) (Leclère et al.,
2016) to obtain a first rough speed estimate. The benefit of MOPA is its ease of use as well as its ability to deal with strong
speed variations without the need for fine-tuning a band-pass filter. Once the initial speed estimate is obtained, this estimate is
refined through the multi-harmonic demodulation (MHD) method which does need an initial rough speed estimate in order to
work properly (Peeters et al., 2022). However, MHD does typically produce much more accurate speed estimates that are on
par with a physical angle encoder. Afterwards, the estimated speed needs to pass a quality check to ensure it actually improves
the ensuing processing steps and will lead to meaningful condition indicators.

Once the instantaneous angular speed is known, the data is angularly resampled and can then be used for further data
cleaning steps. Typically, the data is separated into deterministic and stochastic signal content since gears and bearings are
considered to primarily produce only one of these two distinct signal characterizations. Common methods to achieve this separation are cepstrum editing (Peeters et al., 2018b, 2017), Discrete/Random Separation (Peeters et al., 2020), Linear Prediction
Coding (Antoni and Randall, 2004), Self-Adaptive Noise Cancellation (Ho, 1999), and Phase Editing (Barbini et al., 2017).
This approach does give rise to a tripling of the signals to use for condition indicator calculation as there is now the raw
signal, and the deterministic and stochastic signals. Another commonly employed pre-processing technique is the usage of a
filterbank (Antoni, 2021). Band-pass filtering the signal prior to indicator calculation increases the sensitivity of the computed
statistics to frequency-localized phenomena. Since a fault might be amplified by the transfer path from the source to the receiving sensor, resonances can play an important role in the detection through statistics of the fault (Randall, 2021). Hence adding
frequency-dependency to the condition indicator calculation can greatly enhance the efficacy with which a pipeline is capable
of early fault detection. One of the most popular examples of this aspect is the Kurtogram (Antoni, 2007), which employs a
binary-ternary filterbank to track the kurtosis of different frequency bands. Similar filterbank structures can also be used for
other statistics than kurtosis (Peeters et al., 2019c).

After speed estimation and data cleaning, the final signal processing step is to compute condition indicators on the pre-processed data. For complex machinery, there are usually a lot of potential components that can fail, meaning that a very targeted approach is often not possible due to a lack of historical insights. Therefore, the most common approach is to calculate a wide array of indicators that look at all potential changes in a signal, be it in the time or the frequency domain. In the time domain, several statistics are calculated on all the pre-processed signals, i.e. the signals after deterministic/stochastic separation and filtering. Most of these indicators are common vibration analysis condition indicators such as RMS, kurtosis, crest factor, negentropy, etc. These indicator types can generally be characterized as quantifying either the Gaussianity or the stationarity of a signal (Antoni and Borghesani, 2019; Kestel et al., 2023). In the frequency domain the condition indicators are linked to the characteristic frequencies linked to the kinematics of the drivetrain. The harmonics produced by the gears and shafts are tracked in both the autopower spectra and the envelope spectra to check for increases in first- and second-order cyclostationarity that could be related to degradation (Napolitano, 2016). Also, the sidebands that surround the fundamental harmonics are tracked as they are often a useful indicator for degradation in case of gear wear (Zhang et al., 2021). Due to the pre-processing and the computation of many different features, the total number of condition indicators typically ends up being in the hundreds or even thousands for a single sensor. Multiply that number by the number of sensors and machines and it becomes clear that manual investigation of each indicator trend very quickly becomes completely unfeasible for a human. The difficulty however is dealing with these indicators in an automated manner that allows for early fault detection while also avoiding too many alarms or false positives which would again increase the manual investigation work required. The next sections detail the proposed approach to handle such large sets of condition indicators in a reliable manner on a fleet-wide level.

## 2.2 Normal Behavior Models

Normal behaviour or anomaly detection models are specialized in identifying data observations that deviate from the established pattern of normal behaviour. NBMs are used in applications to identify abnormalities or outliers, such as fault detection, fraud detection, intrusion detection, and medical diagnosis (Chandola et al., 2009). It is crucial to define the normal behaviour of the model to effectively detect anomalies. If the model is trained on data containing abnormalities, it may face difficulty identifying anomalies. The proposed method relies on utilising exclusively healthy data, ensuring that it does not incorporate faulty observations to detect mechanical faults.

The NBMs are trained using historical healthy data collected from sensors installed on wind turbine components. The proposed hybrid approach uses vibrational signals and SCADA data. Figure 2 illustrates that the trained NBMs take the unlabeled indicator as input and label them with healthy, warning, and faulty states at each timestamp. The condition indicators are derived from the vibration signal, while SCADA data is used together with the condition indicators for machine learning models. The NBMs learn the normal behaviour of wind turbine components from the healthy data, and faults are detected when the model predictions observe deviations from the learned normal behaviour. The wind turbine operates in complex weather and environmental conditions, which presents a greater challenge for fault detection methods due to varying wind speeds and changing operating conditions. A model trained on observations from a low wind speed operating regime could misclassify an

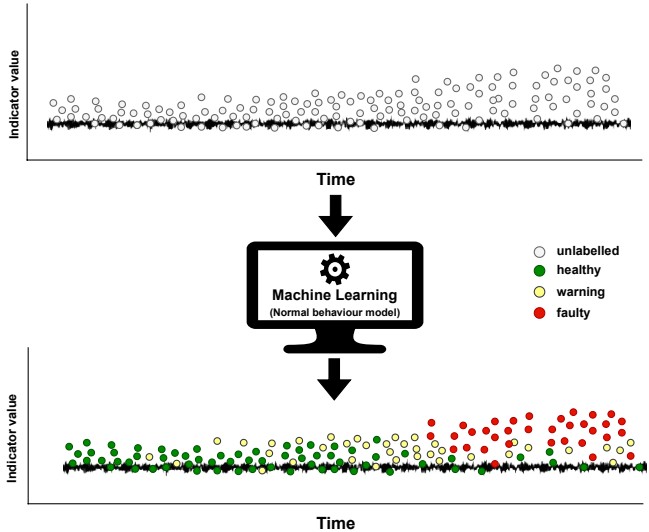

**Figure 2.** The normal behaviour machine learning models label the unlabelled indicators to healthy (green), warning (yellow), and faulty (red) states.

observation as a fault when applied to observations from a high wind speed operating regime. The proposed method addresses this issue by incorporating a step to ensure operating condition independence, to mitigate the influence of varying operating regimes. K-means clustering is used to segment the measured data into distinct operating regimes using wind turbine operational data. The operating parameters from the SCADA data, such as active power and rotation speed are utilised to define these operating regime clusters. Before NBM predicts the health status of the associated indicator, the K-means clustering

model, trained on the operating parameters of healthy data, assigns each observed data point to a corresponding operating regime. An individual NBM is trained for each indicator per operating regime. The objective of a machine learning model is to predict the value of a condition indicator based on the SCADA operating parameters. The model is trained on healthy data to predict indicator values reflecting normal behaviour based on the input SCADA operating parameters. A fault introduces changes in the vibration signal, causing deviations from the expected normal behaviour. As a result, the trained model can

identify faults by comparing the difference between the predicted indicator value, based on the SCADA operating parameters, and the actual measured value. The active power and rotation speed are SCADA operating parameters which serve as the input features for the machine learning model to predict the expected value of a specific signal processing indicator target variable. Various regression models, including linear regression, bayesian ridge regression, support vector regression, multilayer perceptron regression, and decision tree regression, are assessed as NBM. Among these, bayesian ridge regression produced superior

results. Consequently, it is selected for NBM, which is a bayesian approach to linear regression with ridge (L2) regularization. It leverages probability distributions for the regression coefficients to estimate the model parameters and quantify uncertainty in predictions. This probabilistic framework is valuable for robust modelling and assessing the level of uncertainty associated

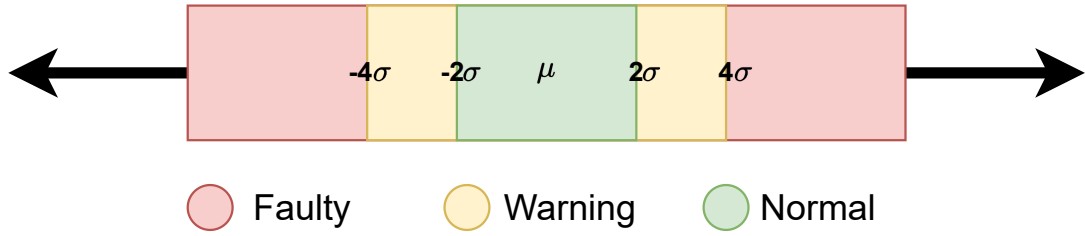

**Figure 3.** A measurement is considered healthy when the difference between the measured and predicted indicator values falls within $\pm 2$ standard deviations, labelled as a warning when the difference is between 2 and 4 standard deviations, and classified as faulty when it exceeds 4 standard deviations.

with the model's coefficient estimates and predictions. Equation 1 represents the Bayesian ridge regression model:

$$Y = \beta_0 + \beta_1 X_1 + \beta_2 X_2 + \epsilon \tag{1}$$

In this equation, $Y$ represents the condition indicator, the variable to be predicted. $X_1$ and $X_2$ correspond to the SCADA operational parameters as active power and rotation speed are the input variables or features of the model. The coefficients $\beta_0$, $\beta_1$, and $\beta_2$ are model parameters, with $\beta_0$ denoting the intercept term. In Bayesian Ridge Regression, these coefficients are considered as random variables, enabling probabilistic modelling. The term $\epsilon$ represents the error term, accounting for data variability and noise. The NBM trained on healthy data, predicts the indicator value corresponding to a healthy machine behaviour based on the provided SCADA operational parameters. The machine's health status, specific to each condition indicator within its operating regime, is assessed by comparing the actual and predicted indicator values using the two-sigma rule, as illustrated in Figure 3. Under this rule, deviations within two standard deviations of the predicted value are considered healthy, capturing approximately 95% of the expected variation in a normal distribution. Deviations between two and four standard deviations are labelled as warnings, while those exceeding four standard deviations are classified as faults. The proposed thresholding approach enables early fault detection to provide operators more time to plan and implement maintenance strategies. However, as sensitivity requirements may vary across applications, an extended $n$-sigma framework can be employed, setting the threshold at $n$ standard deviations from the mean, to adapt the fault detection strategy to specific operational needs.

### 2.3 Alarming

While the individual labelled indicator trends are easy to interpret, observing a large number of indicator trends is challenging and quickly becomes entirely unfeasible for a large number of machines and sensors. The proposed method incorporates an alarming step to aggregate multiple indicators into a single high-level indicator. It provides a comprehensive global health status overview of wind turbine components, eliminating the need to inspect individual indicators separately. The high-level indicator is generated by applying a sliding window across the entire time series of each indicator to identify healthy, warning, and faulty intervals, as illustrated in the Figure 4. A faulty interval represents consecutive detected faulty windows. A sliding window spanning 60 days is utilized, as measurements are taken at intervals of ten seconds approximately every two or three

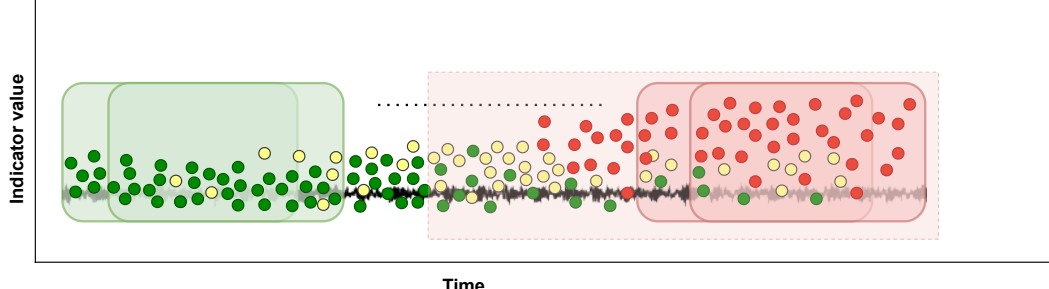

**Figure 4.** The sliding window over a single indicators time series to identify continuous faulty intervals based on labelled as a warning and healthy measurements determined by the NBM.

days. Each measurement is assigned a value of 0, 0.5, or 1, corresponding to healthy, warning, or faulty predicted states by NBM, respectively. The status of each window is determined as healthy or faulty by counting the number of warning and faulty measurements based on these assigned values. The mean of the assigned values is calculated, and if the mean is equal to or greater than 0.3, the window is labelled as faulty. To eliminate outliers, any faulty interval shorter than one month is excluded.

As shown in the Figure 5, the high-level indicator is an aggregation of all indicators computed from a single sensor's vibration signal, providing a high-level health status overview of the components on which it is installed. This high-level indicator depicts the indicators observing faulty intervals. The high-level health indicator is derived by adding the number of indicators having a labelled faulty interval during a month. The high-level indicator is visually represented as a bar plot, showcasing the number of signal processing indicators that are observing faults during one month. Early-stage faults are initially detected by a limited set of signal-processing indicators. As the fault's severity progresses more indicators start observing the fault trend. Furthermore, our proposed method allows experts to examine individual indicators tracking fault trends, facilitating a more detailed examination of the identified faults.

## 3 Experiments

The proposed method is validated on a wind farm comprising more than 50 wind turbines. For the analysis, data is collected at a sampling frequency of 20 kHz using strategically positioned accelerometers on the wind turbine drivetrain components. This data collection process involves measuring data for ten seconds (10s) during a single measurement every two to three days over several years. Consequently, an approximate total of 150 measurements are obtained each year. From these individual raw vibration signal measurements, multiple signal processing statistical and frequency domain indicators are derived. The signal processing indicators are computed from the raw vibration data obtained by a single sensor. The variability in wind speeds and its influence on the wind turbine's operating conditions, the data is segmented into four distinct operating regimes before the NBM training phase. A distinct NBM is trained for each computed indicator per operating condition to label the indicators with healthy, warning, or faulty state at each timestamp. Approximately one year of healthy data is used to train the

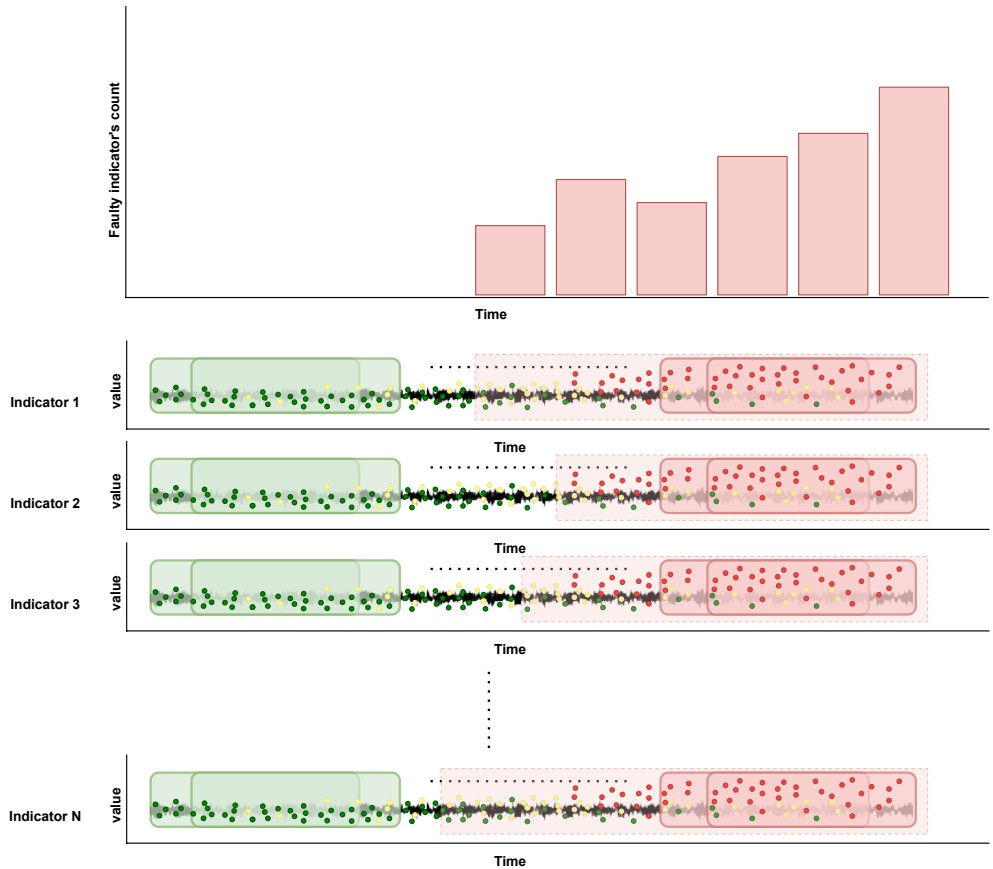

**Figure 5.** The high-level health indicator is derived by combining data from various indicators computed from a single sensor's raw vibration signal. It provides the idea about the number of indicators detecting fault trends at a timestamp.

NBMs, ultimately enabling the ability to track indicator's fault detection trends over multiple years. The labelled indicators are combined into a single high-level health indicator for each sensor located on a specific wind turbine drivetrain component at a particular position. The detected faults by the proposed method are subsequently confirmed via manual borescope inspections conducted by engineers on the drivetrain components.

## 4 Results

The proposed method is validated on an entire offshore wind farm. Due to confidentiality constraints, specific information about the wind farm or individual wind turbines cannot be disclosed. Consequently, plots have been generated with anonymized axes to showcase the results while preserving the confidentiality of the sensitive information. However, for the purpose of the result demonstration, a detailed analysis of four specific cases is presented in this result section. These cases include three instances of fault detection at different wind turbine life stages and one where the wind turbine remains consistently healthy throughout the

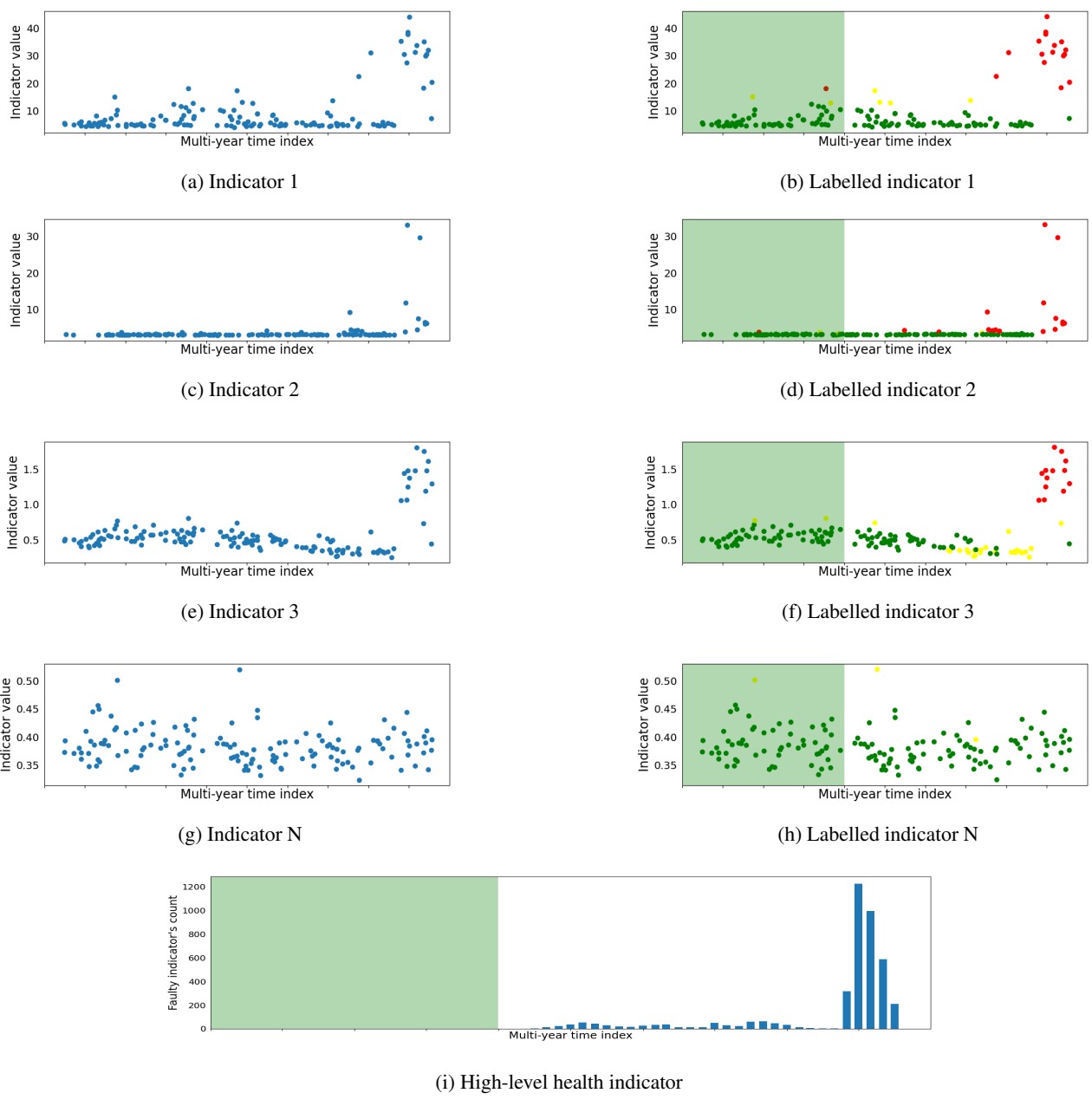

**Figure 6.** A planetary stage channel fault is detected by multiple low-level signal processing indicators and then also depicted in the high-level health indicator. The fault is later confirmed in a comprehensive borescope inspection.

observation period. The result section elaborates on these diverse fault cases associated with different wind turbine drivetrain components to offer a comprehensive illustration of the proposed method's ability to detect faults within rotating components

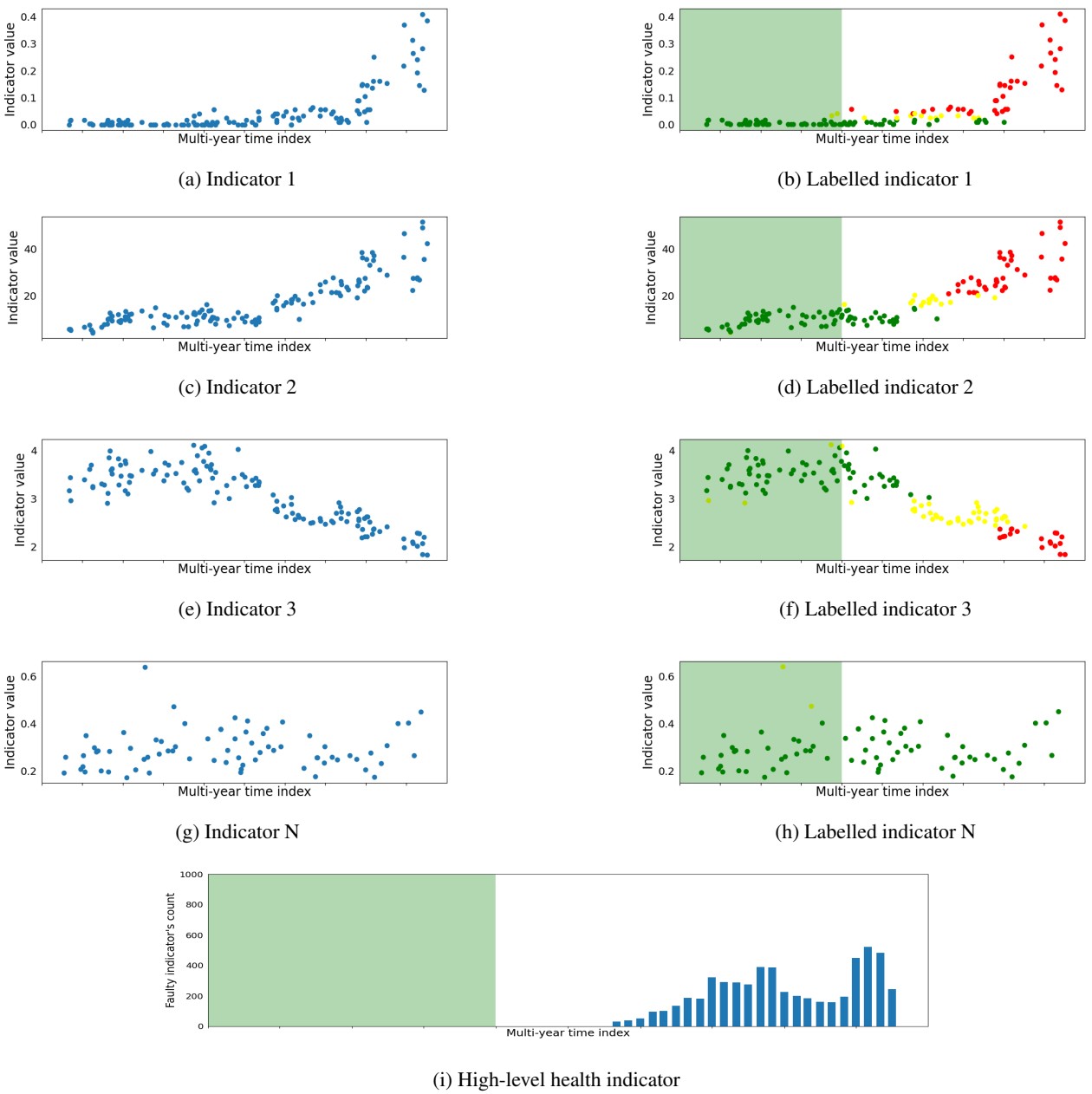

(a) Indicator 1

(b) Labelled indicator 1

(c) Indicator 2

(d) Labelled indicator 2

(e) Indicator 3

(f) Labelled indicator 3

(g) Indicator N

(h) Labelled indicator N

(i) High-level health indicator

**Figure 7.** A generator channel fault is detected by multiple low-level signal processing indicators and then also depicted in the high-level health indicator. The fault is later confirmed in a comprehensive borescope inspection.

of the drivetrain. Figure 6, Figure 7, and Figure 8 depict the faulty scenarios, while Figure 9 illustrates a healthy case. These visual representations showcase the results obtained at three distinct stages of the proposed method. Each figure demonstrates

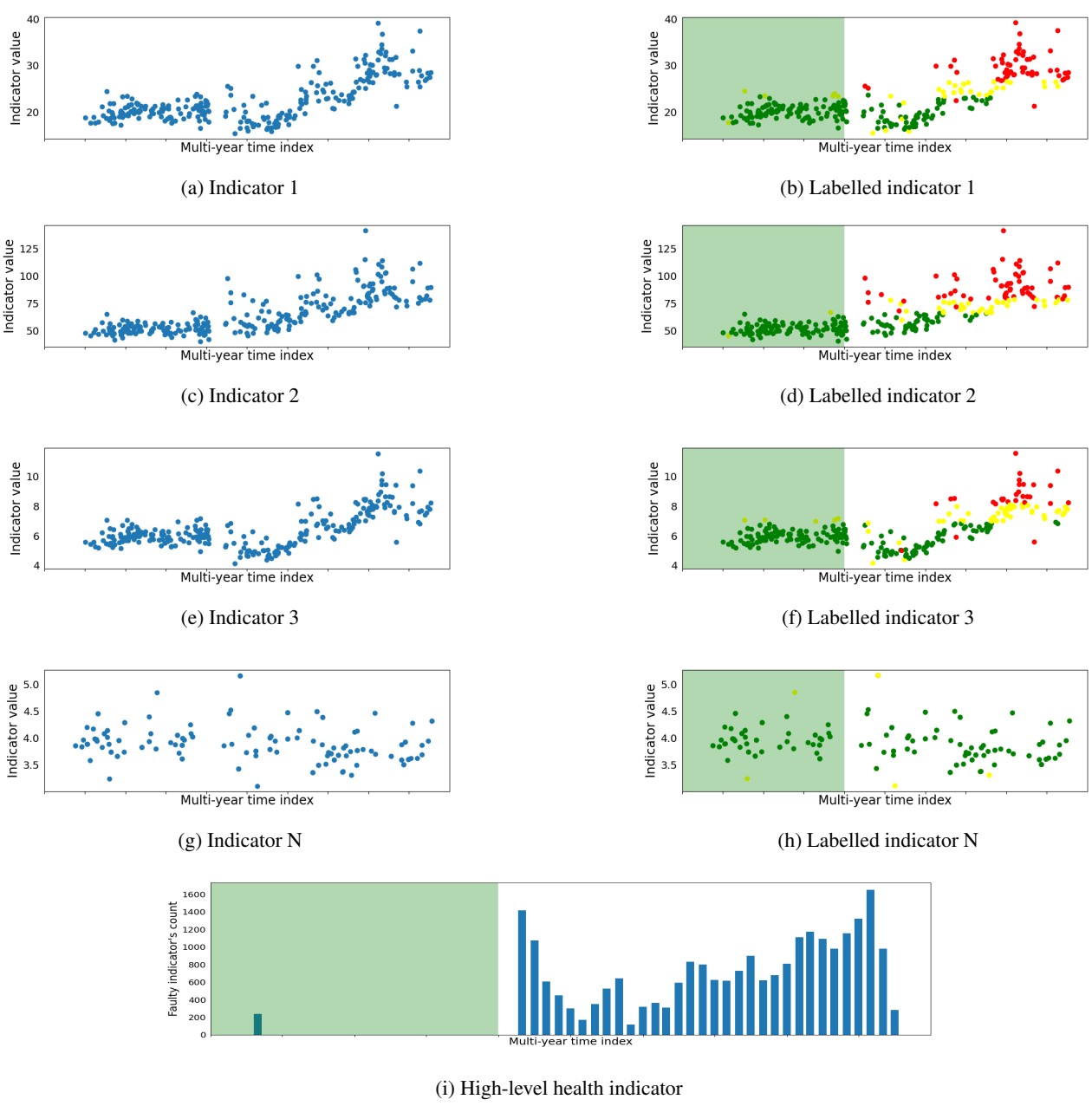

(a) Indicator 1

(b) Labelled indicator 1

(c) Indicator 2

(d) Labelled indicator 2

(e) Indicator 3

(f) Labelled indicator 3

(g) Indicator N

(h) Labelled indicator N

(i) High-level health indicator

**Figure 8.** A high-speed stage channel fault is detected by multiple low-level signal processing indicators and then also depicted in the high-level health indicator. The fault is later confirmed in a comprehensive borescope inspection.

the proposed method output at three different stages, including the derived signal processing indicators, the machine-learning labelled indicators categorized as healthy, warning, and faulty, as well as a high-level health status overview illustrating the

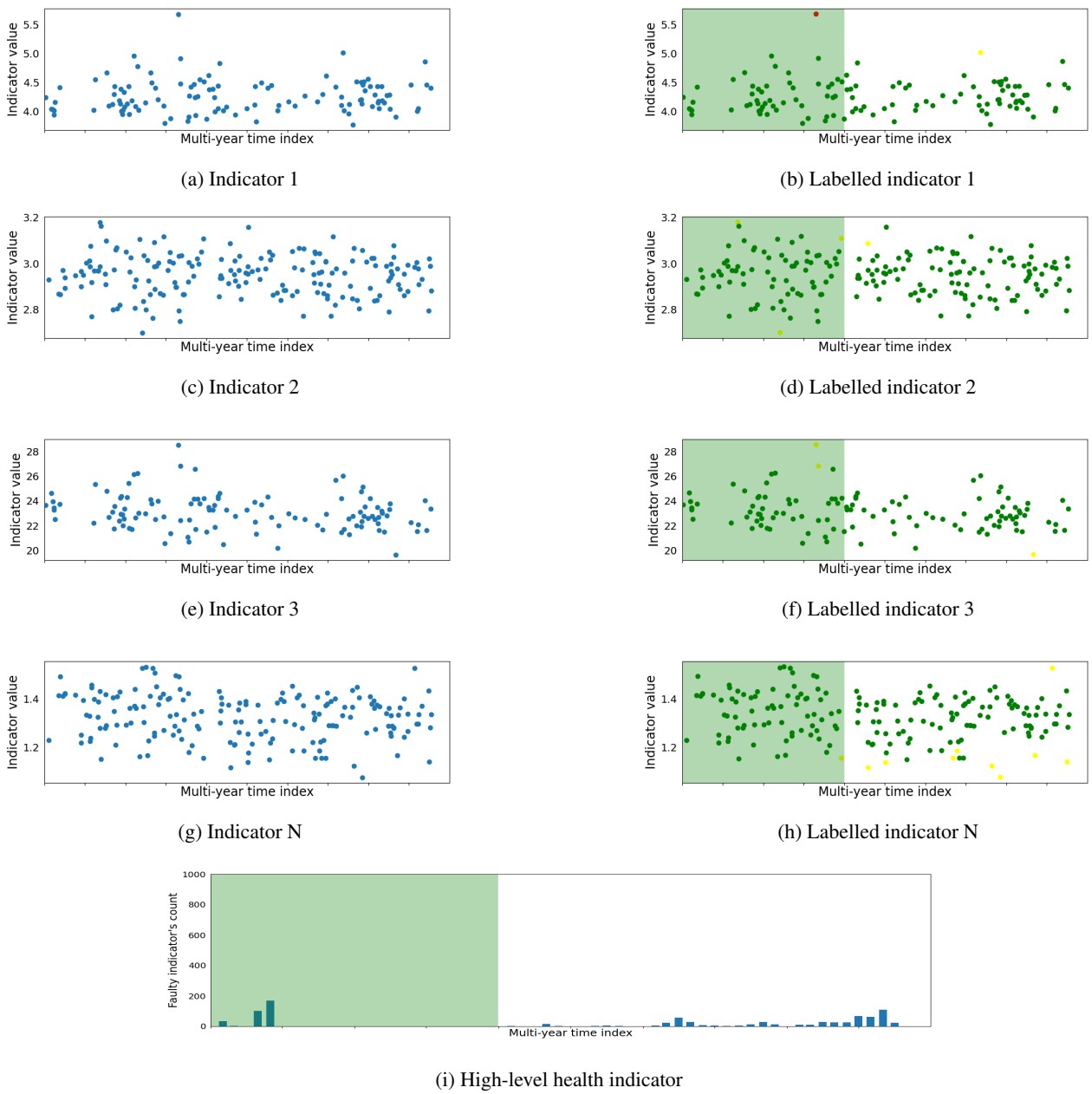

(a) Indicator 1

(b) Labelled indicator 1

(c) Indicator 2

(d) Labelled indicator 2

(e) Indicator 3

(f) Labelled indicator 3

(g) Indicator N

(h) Labelled indicator N

(i) High-level health indicator

**Figure 9.** A planetary stage channel healthy case depicted a healthy state throughout the observed time period.

counts of faulty indicators. The x-axis represents the time of observation spanning multiple years, while the y-axis displays the
290 values of indicators in both labelled and unlabeled plots. Additionally, the signal processing labelled indicators, determined
by the machine learning NMBs, indicate the health status of the indicator at a specific timestamp. It is denoted by green,

yellow, and red colours for healthy, warning, and faulty states, respectively. In the high-level health indicator plot, the y-axis represents the count of indicators observing fault trends. The green shaded area on the labelled and high-level health indicator plots represents the training healthy period for NBMs. Although the green area spans over two years in the figure, it's important to note that a substantial number of measurements are missing during the initial year. Consequently, the actual healthy data measurements are approximately equal to one year.

The planetary stage channel fault is depicted in Figure 6. It is challenging for signal processing indicators to detect early-stage planetary stage channel faults. Therefore, a notable impulsive increase in the number of indicators observing faults is observed instead of a gradual growth in indicators fault trends. The manual inspection revealed indentations on the planet gear and ring gear teeth. Additionally, standstill mark damages are observed on the rollers of the planet gear bearing. Figure 7 illustrates a generator channel fault, observing a gradual increment in the number of indicators detecting fault trends in proportion to the fault's intensity. In the early stages of the fault, only a handful of indicators observe the fault, but as the fault progresses, an increasing number of indicators begin to depict fault trends. Notably, the early detection of the generator channel fault, along with the incremental rise in the number of indicators observing faults over time, allows wind farm operators more time for maintenance planning, as compared to the immediate surge in indicators observed for the planetary stage channel fault. The high-speed stage (HSS) channel fault case is shown in Figure 8, where a significant number of indicators commenced detecting fault trends at the early stage of the fault. This early response signals the introduction of a fault, which engineers confirm through manual inspection, which revealed abrasive wear on the roller flange of the generator-side HSS bearing. In contrast to faulty cases, the case of a healthy planetary stage channel is depicted in Figure 9. The signal processing indicators do not detect any significant fault trends. However, a few indicators exhibit minor fault observations during the typical run-in period when the moving mechanical components are still settling. Furthermore, a minimal fault trend is observed towards the end, but it lacks significance compared to the faulty cases. The healthy case is also verified through a manual inspection, where all bearings and gears exhibit no signs of wear or damage and are found in good condition.

## 4.1   Performance analysis

The proposed method is validated using real wind farm data, which presents challenges in performance analysis due to data imperfections. A key challenge is the uncertainty in the exact timing of fault initiation; however, fault cases are confirmed through manual inspection. Therefore, fault cases identified during a manual inspection are considered actual faults, while the remaining cases are classified as healthy. Additionally, fault-related frequencies may appear in channels monitoring neighbouring components, which further complicate the fault detection. Since precise details about real wind farm data are unavailable, the following assumptions are made to define True Positives (TP), True Negatives (TN), False Positives (FP), and False Negatives (FN):

- TP: The model correctly identifies a manually confirmed fault as faulty.

- TN: The model correctly classifies a healthy case as healthy.

- FP – False Alarm: The model incorrectly predicts a healthy case as faulty.

– FN – Missed Fault Detection: The model fails to detect a fault and incorrectly classifies it as healthy.

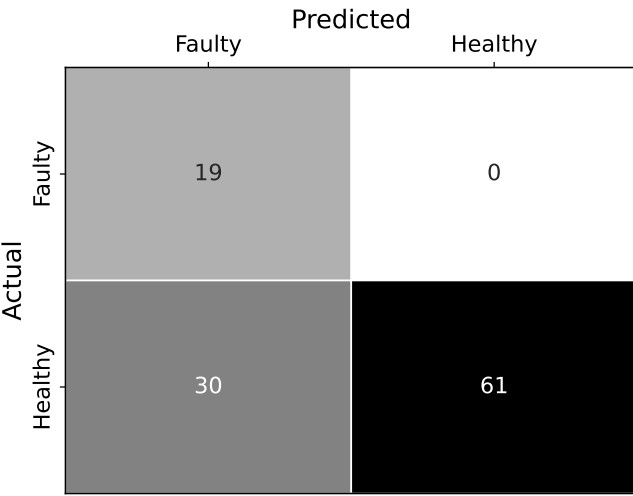

**Figure 10.** Confusion matrix of the fault detection method with 19 true positives, 0 false negatives, 30 false positives, and 61 true negatives.

The performance of the proposed method is evaluated on 10 wind turbines, where manual inspection confirmed faults in 19 drivetrain components of 8 turbines. A confusion matrix is created to evaluate the performance of the proposed method based on the assumptions defined for TP, TN, FP, and FN. The confusion matrix, as shown in Figure 10, indicates 19 TP, 0 FN, 30 FP, and 61 TN. The method successfully detects all faults confirmed during manual inspection and does not predict confirmed

faults as healthy. However, there are several reasons for false-positive predictions. An early-stage fault may not be confirmed during manual inspection, and the channel monitoring a healthy component might register fault frequencies from neighbouring components. The validation dataset consists of data from 10 wind turbines, with 19 confirmed faulty drivetrain components identified through manual inspection, while the remaining 91 channels are considered healthy. Due to the data imbalance, the model's performance cannot be effectively evaluated using standard precision, recall, and F1-score metrics. Instead, weighted

precision Equation 2, recall Equation 3, and F1-score Equation 4 are calculated to provide a comparative performance assessment. Where $N_f$ and $N_h$ are the number of actual faulty and healthy cases respectively. Similarly, $Precision_f$, $Recall_f$, and $F1_f$ represent the precision, recall, and F1-score for the faulty label, respectively. On the other hand, $Precision_h$, $Recall_h$, and $F1_h$ represent the precision, recall, and F1-score for the healthy label, respectively.

$$\text{Weighted Precision} = \frac{(N_f \times \text{Precision}_f) + (N_h \times \text{Precision}_h)}{N_f + N_h} \tag{2}$$

$$\text{Weighted Recall} = \frac{(N_f \times \text{Recall}_f) + (N_h \times \text{Recall}_h)}{N_f + N_h} \tag{3}$$

$$\text{Weighted F1-Score} = \frac{(N_f \times \text{F1}_f) + (N_h \times \text{F1}_h)}{N_f + N_h} \tag{4}$$

Table 1 presents the evaluation metrics—precision, recall, and F1-score—for faulty and healthy cases, along with their weighted averages. The results show that the model achieves high recall (1.00) for faulty cases, ensuring all confirmed faults are correctly detected. For the healthy case, the model demonstrates perfect precision (1.00), meaning all predicted healthy cases are healthy. However, the recall of 0.67 for the healthy class indicates that some actual healthy cases are misclassified as faulty. The

| | Precision | Recall | F1-score |
|---|---|---|---|
| Faulty | 0.39 | 1.00 | 0.56 |
| Healthy | 1.00 | 0.67 | 0.80 |
| Weighted average | 0.89 | 0.73 | 0.76 |

**Table 1.** Weighted evaluation metrics of fault detection method.

weighted precision (0.89) indicates that the model's prediction, across both healthy and faulty cases, is 89% correct. However, the weighted recall (0.73) suggests that the model correctly identifies 73% of actual healthy and faulty cases. Ideally, both precision and recall should be high, but in practice, there is a trade-off between false positives (FP) and false negatives (FN). Increasing recall may reduce false alarms (FP) but could lead to missed fault detection (FN). Balanced accuracy, as defined in Equation 5, is a performance metric used to evaluate a model's accuracy when dealing with imbalanced datasets. The model has achieved a balanced accuracy of 84%, highlighting its reliability in accurately predicting fault cases.

$$\text{Balanced Accuracy} = \frac{1}{2}\left(\frac{TP}{TP+FN} + \frac{TN}{TN+FP}\right) \tag{5}$$

For effective fault detection, it is crucial to minimize false fault alarms while ensuring no fault detection is missed. Moreover, real wind farm data validates the method's real-world applicability but presents challenges, since only faulty cases are labelled with certainty. Therefore, an accurately labelled dataset is essential for precisely evaluating the performance of the proposed method.

## 5  Discussion

The proposed hybrid method combines physics-based signal processing indicators with machine learning techniques to detect faults in wind turbine drivetrains. The labelled signal processing indicators, which include healthy, warning, and faulty states, can be easily analysed without expert knowledge. Furthermore, the high-level health status provides an overview of wind turbine components' health without requiring the inspection of individual condition indicator trends. This proposed method can serve both engineering experts and non-expert operators, offering them a health status overview tailored to their needs. A non-expert operator can gain a high-level understanding of the fleet's health to plan upcoming maintenance campaigns, while an expert can obtain a more detailed overview to understand the nature of the fault and determine the specific maintenance

requirements for a particular component. The method has been validated across the entire wind farm drivetrain components, with borescope inspections confirming the results. For demonstration purposes, we discuss four cases in this study, where data has been observed over multiple years. One year of healthy data is used to train the NBMs in the training phase. The drivetrain components observe the faults at different stages of their lifecycle. Early fault detection of the planetary stage gears and bearings is often particularly challenging and typically leads only to minor increases in condition indicator trends which are difficult to detect manually. The proposed hybrid pipeline accurately tracks the degradation of the planet gear, ring gear teeth, and planet gear bearings rollers faults through the high-level alarm trend. In the remaining two cases, the generator and HSS channel faults are identified at an early stage, and an increasing trend is observed in the high-level indicator. The fault trends observed by the proposed hybrid method are confirmed after monitoring the abrasive wear on the roller flange of the generator-side HSS bearing. In contrast, the planetary stage in the healthy case remains fault-free throughout the observing period, and the inspection confirms no wear and damage on the bearings and gears. The method's performance is evaluated through a study of 10 wind turbine datasets, which are monitored by 110 sensors, including 19 confirmed faulty cases. These faults are confirmed through manual mechanical inspections conducted by technicians. Due to the imbalanced nature of our fault data, balanced accuracy is used as the performance metric. The model has achieved a balanced accuracy of 84%. The proposed method demonstrates high reliability in fault detection, accurately identifying all 19 drivetrain fault cases in the performance analysis of 10 wind turbines. However, some false alarms arise due to the complexities of real wind farm data. A high threshold reduces false alarms but increases the risk of missing actual faults. Therefore, a lower threshold is adopted to prioritise early-stage fault detection, as identifying potential issues as early as possible is critical to provide sufficient time to plan and execute maintenance strategies. A higher threshold may reduce false alarms, but it can compromise the detection of subtle changes in indicator trends, limiting the ability to identify faults at an early stage. Additionally, it becomes challenging to suppress false alarms when a sensor captures fault frequencies originating from neighbouring components.

The proposed hybrid method effectively provides a comprehensive assessment of the turbine's health, encompassing both a high-level overview and detailed insights into individual condition indicators for each installed accelerometer. However, the training process demands substantial computational resources due to the requirement of individually training an NBM for each condition indicator. Future work aims to address this challenge by developing an explainable machine learning model that can adapt to all condition indicators simultaneously while providing both a holistic evaluation of the turbine's health and an in-depth analysis of individual condition indicator fault trends. It will significantly mitigate the computational burden. The multitude of condition indicators employed in this approach plays a critical role, as each indicator demonstrates sensitivity to distinct fault categories. By leveraging the correlations among indicators, it becomes feasible to streamline the number of necessary indicators while maintaining robust fault detection capabilities. Nevertheless, this requires a detailed data analysis of indicators over multiple cases before eliminating any indicators. The proposed method integrates two data sources: the vibrational signals measured by accelerometers installed on the components and SCADA data. As a consequence, the application of this hybrid condition-monitoring fault detection method is limited to wind turbines that are equipped to measure both vibration and SCADA data. Future work will focus on vibrational analysis to reduce dependence on SCADA data. However, develop-

ing a standalone vibration-based condition monitoring method will require high-quality vibration data to extract operational information.

## 6 Conclusions

A hybrid fault detection method was introduced that combines advanced signal processing techniques with machine learning to offer a comprehensive overview of the health of wind turbine drivetrain components. The proposed method provided a high-level health status overview to address the vast number of condition monitoring indicators, as individually monitoring all available condition indicators is not possible when managing multiple components in a single wind turbine across an entire wind farm. This method not only facilitates high-level health assessments but also allows for in-depth inspections of signal processing indicators, making it a versatile tool suitable for both experts and non-expert stakeholders. The proposed method has been validated across an entire wind farm fleet, where it consistently achieved satisfactory results in the majority of cases. To further validate its effectiveness, manual borescope inspections were conducted after the fault detection process, confirming the presence of mechanical faults. The combination of physical knowledge and the computational power of machine learning in our approach holds great promise for enhancing the reliability and efficiency of wind turbine maintenance and performance monitoring.

*Author contributions.* FJ was responsible for conceptualization, data curation, formal analysis, investigation, methodology development, project administration, software coding, validation, visualization, and writing of the original draft. CP contributed to conceptualization, methodology development, project administration, supervision, and the reviewing and editing of the manuscript. TV was involved in conceptualization, methodology development, and supervision. JH was responsible for conceptualization, funding acquisition, methodology development, supervision, and the final phase of paper review and editing.

*Competing interests.* The contact author has declared that none of the authors has any competing interests

*Acknowledgements.* The authors would like to acknowledge FWO (Fonds Wetenschappelijk Onderzoek) for their support through the SB grants of Faras Jamil (#1S63123N), post-doctoral grant of Cédric Peeters (#1282221N), and SBO project Robustify (S006119N). Furthermore, this research was supported by funding from the Flemish Government under the "Onderzoeksprogramma Artificiële Intelligentie (AI) Vlaanderen" programme. The authors are also grateful to the VSC Supercomputing Flanders centre for the support in the context of the VSC Cloud program.

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
