# Peer review of "Leveraging Signal Processing and Machine Learning for Automated Fault Detection in Wind Turbine Drivetrains"

_Wind Energy Science, 2024_

## Author Response (AR1)

**Author's response**

**RC1 (Reviewer 1):**

We thank the Reviewer for taking the time to provide detailed feedback on our manuscript. Following are the responses to the reviewer feedback. The structure is as follows: 1) **Feedback** from the reviewer, 2) **Response** to the feedback, and 3) **Changes to the manuscript:** based on the latexdiff file.

**Feedback:**
**Clarity of Presentation:** Figures, especially 6–9, are vital for understanding the results but are difficult to interpret due to anonymized axes and scales. While confidentiality is necessary, providing normalized or generalized labels (e.g., "Normalized Time," "Fault Indicator Count") would make the visualizations more accessible without compromising sensitive data. Additionally, the Bayesian Ridge Regression model is only briefly introduced; a more detailed justification for its use compared to other regression techniques would improve the methodological section.

**Response:**

The data used to validate the method is private, and we are not allowed to share specific details regarding the machine configurations or the monitoring period. Therefore, anonymised axes are used to present the results. However, as highlighted in the feedback, the high-level health indicator in each result figure (specifically the (i)th subfigure) shows the fault indicator and counts the number of condition indicators that are exhibiting a fault trend at a given time.

Various regression techniques were evaluated before selecting Bayesian Ridge Regression. The details regarding the comparison of different regression models are now included in the manuscript, specifically in the Normal Behavior Models subsection.

**Changes to the manuscript:**

Page: 8 Lines: 223-226 in Normal Behavior Models subsection.

**Feedback:**
**Technical Gaps:** While SCADA data is mentioned as an input, the integration process with vibration data is underexplored. Elaborating on preprocessing steps or synchronization challenges would provide a more comprehensive picture. Moreover,

the paper does not report performance metrics such as precision, recall, or false alarm rates for the NBMs or hybrid system. Quantitative validation is crucial to assessing the method's practical value.

**Response:**

As we are using real wind farm data, determining the exact time a fault is introduced into the system is challenging, making it difficult to provide a precise quantifiable performance measurement of the method. However, as suggested by the reviewer, we have conducted a performance analysis on data from 10 wind turbines, based on the condition monitoring report and fault cases confirmed by technicians through manual inspection. This study provides a more quantitative assessment of fault prediction, including evaluation metrics such as precision, recall, and F1-score.

The integration of SCADA data and vibration data is included in the "Hybrid Condition Monitoring Fault Detection Method" section.

**Changes to the manuscript:**

Pages: 16-18 Lines: 313-357 in Performance analysis subsection.

Page: 5 Lines:132-138 in Hybrid Condition Monitoring Fault Detection Method section.

**Feedback:**
**Scalability Concerns:** The approach requires training individual NBMs for each condition indicator across different operating regimes, which is computationally intensive and may be impractical for large-scale deployment. Although the paper acknowledges this issue, it does not propose concrete steps to address it. Exploring techniques like transfer learning, ensemble models, or feature reduction could mitigate this limitation.

**Response:**
The current approach offers a detailed analysis of each condition indicator but requires significant computational resources since separate NBMs must be trained for each indicator in every operating regime. To address this challenge, future work will focus on developing an explainable deep learning model that can adapt to all condition indicators simultaneously. This will provide both a high-level assessment of the turbine's overall health and a detailed analysis of individual fault trends while significantly reducing computational demands.

**Feedback:**
**Limited Discussion of Related Work:** The paper could better contextualize its contributions by comparing the proposed method to other state-of-the-art approaches, such as fully data-driven deep learning systems. Highlighting the relative strengths and weaknesses of the hybrid approach would provide a clearer perspective on its novelty and utility.

**Response:**

The related work section has been expanded by adding more information about SCADA-based condition monitoring in introduction section.

**Changes to the manuscript:**

Page: 2 Lines: 50-55 in Introduction section.

**Feedback:**
**Broader Applicability:** The reliance on both vibration and SCADA data limits the applicability of the method to turbines equipped with these systems. The paper would benefit from discussing adaptations for turbines with only partial data availability or exploring how the method might generalize to other industrial systems.

**Response:**

This research focuses on vibration-based condition monitoring and presents a framework for assessing the health of wind turbine drivetrain components using vibration data. While SCADA-based methods exist in the literature, vibration-based approaches are generally more reliable. Additional details on SCADA-based condition monitoring have been incorporated into the introduction section. Moreover, future work will focus on extracting operational information directly from high-quality vibration data to reduce reliance on SCADA data. This update is included in the discussion section.

**Changes to the manuscript:**

Page: 2 Lines: 50-55 in Introduction section.

Page: 19 Lines: 394-396 in the Discussion section.

**Feedback:**
**Results Presentation:** While the paper discusses case studies qualitatively, it lacks numerical summaries or statistical analysis of the detection performance. Providing such data would strengthen the evidence for the method's effectiveness. Additionally, the scalability of the method across a fleet of wind turbines needs to be demonstrated more convincingly.

**Response:**

To quantitatively evaluate the performance of the proposed method, we have added a confusion matrix in the Performance analysis subsection, along with evaluation metrics such as precision, recall, and F1-score, to assess fault detection effectiveness.

**Changes to the manuscript:**

Pages: 16-18 Lines: 313-357 in Performance nalysis subsection.

**RC2 (Reviewer 2):**

We sincerely thank the Reviewer for taking the time to review our research and provide valuable feedback. Below is our response, the structure is as follows: 1) **Feedback** from the reviewer, 2) **Response** to the feedback, and 3) **Changes to the manuscript:** based on the latexdiff file.

**Feeback:**

Would be interesting to see one of these studies beyond simulation.

**Response:**

The proposed method has been validated using real wind farm data, with results presented using anonymized axes due to confidentiality constraints. Additionally, to provide the quantitative evaluation, a performance analysis is conducted on datasets from 10 wind turbines. This analysis includes a confusion matrix, which provides a quantitative assessment of the method.

**Changes to the manuscript:**

Pages: 16-18 Lines: 313-357 in Performance nalysis subsection.

**RC3 (Reviewer 3):**

We thank the Reviewer for providing constructive feedback on our research. Our response is provided below:

**Feedback**:

In the model validation session, the authors only showcase the prediction result in figure 7-9 without further statistics on the results. It is not straightforward to get a sense of the model performance. Hence, could the authors provide a more quantified measurement for the performance of the trained machine learning model? For example, what is the percentage of correctly predicted faults? What is the ratio of false positive and false negative?

**Response:**

As suggested by the reviewer, we have conducted a performance analysis on data from 10 wind turbines, based on the condition monitoring report and fault cases confirmed by technicians through manual inspection. This study offers a more quantitative assessment of fault prediction, including false alarms and missed fault detections.

**Changes to the manuscript:**

Pages: 16-18 Lines: 313-357 in Performance nalysis subsection.

---

## Author Response (AR2)

**Author's response**

**RC4 (Reviewer 4):**

We thank the Reviewer for taking the time to provide detailed feedback on our revised manuscript. Following are the responses to the reviewer feedback. The structure is as follows: 1) **Feedback** from the reviewer, 2) **Response** to the feedback, and 3) **Changes to the manuscript:** based on the latexdiff file.

**Feedback:**
The reported confusion matrix reveals a notable imbalance: 30 false positives and 0 false negatives. While minimizing false negatives is critical in fault detection—particularly in wind turbine drivetrains where undetected faults can escalate into severe mechanical failures—a high number of false positives also poses practical challenges, such as unnecessary maintenance actions and associated costs.

The current model uses a two standard deviation threshold for classifying a measurement as faulty. This leads to the following questions:

- Is this threshold deliberately chosen to prioritize fault sensitivity over specificity?

- Was the trade-off between false positives and false negatives empirically evaluated?

- Would a more conservative threshold (e.g., three standard deviations) significantly reduce false positives without compromising early fault detection?

A brief discussion of the rationale behind the threshold selection, including whether any sensitivity analysis was performed to explore different thresholds, would strengthen the methodological justification and help readers understand the model's design priorities. Even if two sigma remains the chosen setting, explaining the decision will improve transparency and reinforce the paper's practical relevance.

**Response:**

The proposed method employs the two-sigma rule for fault detection. A measurement is flagged as faulty when the deviation between the actual and predicted indicator exceeds two standard deviations from the mean. Additionally, the method incorporates a two-level alarm approach:

1. Deviations between two and four standard deviations are classified as warnings,
2. Deviations exceeding four standard deviations are classified as alarms.

We acknowledge that this was inaccurately described in the originally submitted manuscript, and it has now been corrected in the revised version.

The following are our detailed responses to each of the questions and suggestions raised by the reviewer:

**Is this threshold deliberately chosen to prioritise fault sensitivity over specificity?**

The threshold was chosen to prioritise fault sensitivity over specificity. The primary objective of this study is early fault detection, which depends on the ability to identify subtle changes in indicator trends that may signal the initiation of a fault. These early changes are typically confirmed when the fault trends continue to grow over time. The proposed approach facilitates predictive maintenance by enabling human experts to visually assess indicator trends before initiating physical inspections. This intermediate step helps reduce false alarms by allowing experts to evaluate the evolution of fault indicators over time. In critical systems like wind turbines, such early detection is particularly valuable, as it can significantly reduce unplanned downtime and associated maintenance costs.

**Was the trade-off between false positives and false negatives empirically evaluated?**

An empirical evaluation of the trade-off between false positives and false negatives has not yet been conducted in this study, as the focus was primarily on early fault detection. The proposed method was validated using real wind farm data, which introduces several challenges for such analysis. A major limitation is the uncertainty in the exact timing of fault initiation. Although fault cases were confirmed through manual inspection, the lack of precise information makes it difficult to accurately label data for a robust classification of false positives and false negatives.

Additionally, the complex structure of wind turbine gearboxes means that fault-related frequencies may also appear in sensors monitoring neighbouring components. This further complicates the identification of true vs. false alarms and makes threshold selection particularly sensitive.

Given these constraints, fault cases verified by manual inspection were treated as ground truth, and all other cases were treated as healthy. However, to systematically evaluate the false positive/negative trade-off, a more controlled and reliably annotated dataset would be appropriate.

**Would a more conservative threshold (e.g., three standard deviations) significantly reduce false positives without compromising early fault detection?**

A higher threshold can help reduce false positives; however, it comes at the cost of decreased sensitivity to subtle changes in indicator trends, which are critical for early fault detection. In our case, using a higher threshold risks missing early signs of degradation. Moreover, due to the complexity of the wind turbine gearbox, fault frequencies originating from one component may also be captured by neighbouring sensors. As a result, even a conservative threshold may not be able to eliminate false alarms without compromising the model's ability for early fault detection.

**Changes to the manuscript:**

Page: 9 Lines: 228-230 and 235-245 in Normal Behaviour Models subsection.

Page: 19 Lines: 385-390 in Discussion section.